# Variations in Oppel–Kundt Illusion Strength Among Depressive and Schizophrenia Spectrum Disorder Groups: Impact of Benzodiazepine Use

**DOI:** 10.3390/medicina61050835

**Published:** 2025-05-01

**Authors:** Edgaras Diržius, Rasa Pakanavičiūtė, Deimantė Andriuškevičiūtė, Darius Leskauskas, Aleksandr Bulatov

**Affiliations:** 1Institute of Biological Systems and Genetic Research, Lithuanian University of Health Sciences, LT-44307 Kaunas, Lithuania; aleksandr.bulatov@lsmu.lt; 2Department of Psychiatry, Lithuanian University of Health Sciences, LT-44307 Kaunas, Lithuania; rasa012@gmail.com (R.P.); darius.leskauskas@lsmu.lt (D.L.)

**Keywords:** depression, schizophrenia, benzodiazepines, illusion, visual perception

## Abstract

*Background and Objectives*: The Oppel–Kundt (O–K) geometric optical illusion has been studied among people with mental disorders to understand the differences in their visual perception. Earlier studies were mainly focused on patients with schizophrenia, while less is known about patients with depression and the influence of medication use. The objectives were to compare illusion manifestation for schizophrenia, depression, and to evaluate possible differences depending on drug use. *Materials and Methods*: The stimuli consisted of three horizontally arranged dots, which were considered as terminators specifying the ends of the reference and the test stimulus intervals. The reference interval was filled with a set of distracting dots and changed, at random, from 0 to 19. The participants were asked to place the central terminator in the middle, between the outer ones. The trial consisted of 10 different figures, and each trial was repeated 10 times. This study involved 35 patients with depression and schizophrenia spectrum disorders and a comparison group of 35 persons. Information about drug use by patients was retrieved from their medical records. *Results*: OK illusion manifested stronger in patients with depression compared to the other subjects. The patients who were taking benzodiazepines made greater errors evaluating OK figures than those who were not. No differences were found regarding other drug use. The limitations include a limited sample and possible interfering effects of other drugs, especially antidepressants, which have been shown to affect illusion perception. *Conclusions*: The OK illusion was more prominent in the patients with depression and in those who were taking benzodiazepines.

## 1. Introduction

J. J. Oppel first introduced the illusion of filled space in 1855: a spatial interval subdivided into parts looks longer compared to an undivided interval of the same length. This phenomenon was further investigated by A. E. Kundt in 1863 [1]. There is a wide variety of graphic patterns of various configurations with different visual elements that evoke the Oppel–Kundt illusion (O–K). In recent years, several variants of the illusion have been used in the examination of both healthy and clinical population groups [2,3,4,5,6,7]. Anomalies in the organization of perception have been found in people with schizophrenia, which can be used as a potential biomarker useful in the diagnostic process [8]. The tilt illusion and the Müller–Lyer illusion have even been proposed as possible biomarkers of schizophrenia [9,10]. Most of the studies evaluating the perception of illusion-provoking figures in people with mental disorders have focused on patients with schizophrenia [11,12,13,14,15,16]. Although there is evidence of a decrease in retinal contrast gain and visual contrast sensitivity among people with depression, there is a lack of studies regarding geometrical illusions in these patients [15,16]. It was found that the use of benzodiazepines (BZDs) can cause differences in the manifestation of visual orientation illusions, causing integration difficulties, reduced contrast sensitivity, and prolonged visual information processing [17,18,19,20,21,22,23]. There are no recent studies examining the peculiarities in the perception of O-K figures in people with psychiatric disorders and in psychotropic drug users. The results of previous investigations into the manifestation of O-K figures for patients with neuropsychiatric disorders are not coherent. Some researchers found no difference in the perception of O-K figures in patients with schizophrenia. Others found higher errors in patients with paranoid schizophrenia when compared with simple schizophrenia, but these studies did not include a healthy comparison group, and some studies showed that persons with mental retardation made smaller errors [24]. The manifestation of O-K in patients with hemineglect was investigated, but the results for the comparison group were also not included [7,25]. Although it is known that patients with schizophrenia experience a number of difficulties with perception during the visual processing of O-K figures, there is an obvious lack of research among people suffering from, for example, depression. Therefore, we decided to conduct a more detailed study of the perception of O-K stimuli in patients with depression and schizophrenia [8,10,15,17,18,19,20,22,23,26].

The aim of this study was to investigate the O-K illusion manifestation peculiarities in patients with mental disorders. The objectives were to compare illusion manifestation for schizophrenia and depression and to evaluate the possible differences depending on drug use.

## 2. Materials and Methods

### 2.1. Participants and Design

The sample consisted of patients in the psychiatric clinic of the Hospital of the Lithuanian University of Health Sciences (LUHS), Kaunas Clinics. All patients treated during two consecutive months were invited to participate in this study. The Kaunas Regional Studies Bioethics Committee approved the study protocol (Nr. BE-2-27), and all the participants agreed to take part in this study under their own free will and signed the informed consent form. The exclusion criteria for subjects were neurological or ophthalmological diseases, which could affect visual perception, motor system diseases causing an inability to perform the test, alcohol intake during the period of hospitalization, and a subject’s own refusal. The subjects with myopia were allowed to use prescribed eyeglasses. The patients’ drug prescriptions and diagnoses were collected from their medical documents.

Seventy people were included in this study. The study sample consisted of 35 patients: 18 females and 17 males. The median age of the subjects was 32, with the youngest being 18 and the oldest being 63. The comparison group consisted of 35 matched volunteers in good health (Table 1). Each subject was evaluated with the Brief Psychiatric Rating Scale by a certified psychiatrist. If there was any suspicion of a risk of any kind of mental disorder, the subjects were not included in the comparison group [27].

We assigned patients to two groups: a schizophrenia spectrum disorder group (SSDG) (n = 19) and a depressive disorder group (DDG) (n = 16). Diagnoses were confirmed by two psychiatrists in accordance with the International Classification of Diseases (10th version, Australian modification). The SSDG included patients with diagnoses of schizophrenia, schizotypal, and delusional disorders (F20–F29). The DDG consisted of patients with diagnoses of depressive episode (F32) and recurrent depressive disorder (F33); only patients with an episode of moderate and severe depression without psychotic symptoms were included.

### 2.2. Apparatus

The psychophysical experiments were conducted in a dark room so as to reduce the possible impact of uncontrolled factors. A Samsung S23B350B monitor (Samsung Electronics Co., Ltd., Suwon-si, Republic of Korea) was used for stimulus presentations. A Cambridge Research Systems OptiCAL photometer (Cambridge Research Systems Ltd., Rochester, UK) was used to monitor luminance range calibration and gamma correction. A chin and forehead rest was used to maintain a constant viewing distance of 350 cm (at this distance, each pixel subtended about 0.3 arcmin); an artificial pupil (an aperture with a 3 mm diameter of a diaphragm placed in front of the eye) was applied in order to reduce optical aberrations. Only the right eye was tested. The experiments were conducted under the control of our own-designed computer software, which arranged the order in which the stimuli were presented on the monitor, introduced alterations according to the subject’s command, recorded the subject’s responses, and registered the results.

### 2.3. Stimuli and Procedure

The stimuli used in this study comprised three base dots (dot size, 2 arcmin; luminance, 20 cd/m^2^) arranged horizontally, which were considered as terminators specifying the ends of the left (reference, R) and right (test, T) spatial intervals (Figure 1). In all the experiments, the stimulus length (i.e., the sum of interval lengths, R + T) was fixed at 100 arcmin. The reference interval was filled with a set of equally spaced dots according to the Oppel–Kundt pattern. The initial position of the central terminator dot on the horizontal line between the outer terminators was changed at random (evenly distributed within ±10 arcmin). The subject’s goal was to place the central terminator dot in the middle between the outer ones (with the filling dots being rearranged appropriately). The resulting difference in the length of the spatial intervals of the stimulus was considered as an indicator of the strength of the illusion. The time taken to perform the task was not limited. The individual thresholds for the subjects were calculated in comparison with their responses to stimuli without an illusory effect (i.e., with an empty reference interval). A total of 20 stimulus presentations with randomly distributed values of the independent variable (the number of filling dots) were included in a single experimental run, and each subject repeated the experiment ten times on different days. In the present study, we ignored the possible influence of time (the day of testing) on the magnitude of the illusion; however, previous studies have shown that, for example, the Müller–Lyer illusion tended to decrease somewhat with repeated visual exposure [28,29,30].

### 2.4. Drugs

Benzodiazepine-class drug doses were converted to diazepam equivalents [31]. This standardization process guaranteed the uniformity and consistency of our analysis. We determined the conversion factors based on well-established pharmacological sources and clinical standards, enabling a trustworthy and meaningful comparison of the impact of various benzodiazepines on our study participants. This methodology strengthens the credibility of our results and supports a comprehensive assessment of outcomes associated with benzodiazepine usage. We converted antipsychotics to chlorpromazine equivalents in order to make the comparison easier (Table 1) [32]. All of the subjects in the SSDG received antipsychotics, and 11 of 16 subjects in the DDG received antipsychotic medications. All of the subjects in the DDG received antidepressant medication, while 4 subjects in the SSDG received antidepressant medication. Two subjects in the DDG were prescribed an anticonvulsant, while five subjects in the SSDG were prescribed anticonvulsant medication.

### 2.5. Statistical Analysis

Individual responses to stimuli without an illusory effect (i.e., with an empty reference interval) were subtracted from the data for each subject. For the statistical analysis, we used a one-way ANOVA and post hoc Dunnett’s T3 test to assess the differences between the groups. Significant results were considered when *p* < 0.05. The statistical analysis was performed with IBM SPSS statistics software, version 29.0.0.

## 3. Results

### 3.1. Mental Disorders and O-K Figures

The participants in the DDG made significantly greater mistakes than the other groups when evaluating the O-K figures. In the analysis comparing the control group and the DDG, statistically significant differences (*p* < 0.05) were discovered in the number of dots across all cases. Moreover, in the comparison between the SSDG and the control group, notable differences were observed in the cases where dots were not presented. Furthermore, when contrasting the SSDG with the DDG, there were significant differences in the perception of the number of dots in nearly all the cases (*p* < 0.05), with the exceptions being when there were either no dots or nineteen dots (Figure 2 and Table 2).

### 3.2. Medication and O-K Figures

Six patients were not taking any antipsychotic medication, while others were using some form of such medication. Significant differences when comparing the perceptions of O-K figures between antipsychotic medication users and non-users were not found. The patients also used various other types of medications, including antidepressants, anticonvulsants, and antimuscarinic drugs. Significant differences in the perception of O-K figures in relation to medications other than benzodiazepines were not found.

The benzodiazepine-user group consisted of 17 persons (9 males and 8 females aged 22–53 years). We did not find any dose-dependent correlation with benzodiazepine and visual perception; thus, all benzodiazepine users were included independent of benzodiazepine dose. The benzodiazepine non-user group consisted of 18 persons (8 males and 10 females aged 18–63 years). An analysis was conducted to examine the connection between the use of benzodiazepine medications and the occurrence of the O-K illusion. When the referential interval was not filled with dots, no or minor illusory effects were provoked for either group, and the results did not differ among the patients. When the referential interval was filled with three or more dots, the subjects in the benzodiazepine-user group made more significant errors. Changes in illusion activity among the benzodiazepine users were greater, noting that the subjects made larger errors in their perception of the O-K illusion. The results show that taking benzodiazepine medications can lead to a stronger manifestation of the O-K illusion. The patients with depressive disorders who used BZD made larger errors when evaluating the O-K figures (Figure 3 and Table 3).

## 4. Discussion

The results of our study show that the O-K illusion tended to manifest more strongly in the patients with depression when compared to the patients on the schizophrenia spectrum, those with other mental disorders, and those in the comparison group. We also found that the patients who were using benzodiazepines made larger errors than those in the comparison group and the other patients. We did not identify any relationship with the other drugs used by the patients.

### 4.1. Depression

Our results for depressed patients are consistent with previous studies, which showed that depressed patients have reduced visual contrast discrimination performance [15,16,33]. The stimuli in our study were presented in a high contrast manner, which differs from previous studies on contrast sensitivity [15,16]. Reduced electroretinogram patterns were found for patients with depression [34]. Signals in the electroretinogram are mainly generated by ganglion cells [35]. Retinal dysfunction of contrast processing among major depression patients also appeared in the cortical activity [36]. Projections from retinal ganglion cells transmit to the serotoninergic raphe nuclei, which, in turn, are implicated in a broad assortment of behavioral and physiological functions, including affective and visual perception modulation; this could be a possible mechanism by which visual perception is intertwined with the pathophysiology of depression [37]. It was even shown that reduced contrast gain normalizes after an anti-depressive therapy; however, since ganglion cells are important in various visual functions, these effects could not be exceptional in contrast gain, which is a precedent for further studies [34]. Also, it should be noted that patients with mental disorders showed length misjudgments even when there were no distracting dots present. It is possible that this is a general tendency for people with various mental disorders. If this perceptual shift also takes effect when distractor dots are present, results could show themselves in different manners, which would be worthwhile to investigate. Although further investigation is needed to confirm results, the perception differences we found could be a possible biomarker for depression.

### 4.2. Schizophrenia Spectrum Disorder

We did not find any results suggesting that the patients with SSD who did not receive benzodiazepine medication (schizophrenia spectrum disorders) are susceptible to the O-K illusion in a different manner from the healthy comparison subjects and the patients with other disorders. These results contrast with other studies that found deficits in perceptual organization and visual binding tasks among people with SSD [38,39,40]. Recent studies support the assumption that O-K illusion emergence can be associated primarily with the integration of distractor-evoked effects in regions surrounding the endpoint and visual attention focus parameters. However, the illusions investigated may also be associated with several different mechanisms [41,42].

### 4.3. Benzodiazepines

The results reflect a possible effect of benzodiazepine use in strengthening the O-K illusion manifestation for patients with mental disorders. Though it is difficult to provide an explanation regarding the mechanisms of perception differences, our results are in line with other researchers who found that benzodiazepines can cause difficulties in visual orientation, illusion manifestation, visual integration, and attention [17,18,19,22,23,43,44]. The risks of benzodiazepine use are widely known, and further research is needed to better understand the impact of benzodiazepines on visual perception. However, our findings could help in the research of potential biomarkers of benzodiazepine-related cognitive impairment [31].

### 4.4. Study Limitations

Our sample was limited to 54 subjects; it would, therefore, be beneficial to increase the sample size in future studies. Patients in the DDG were treated with an antidepressant medication, but previous studies suggest that antidepressant medication normalizes perception for patients with depression [33]. The difference found in the perceptual performance of the patients taking different doses of treatment drugs is worth exploring further in order to develop a deeper understanding of the effects of medication on visual perception. Also, all the patients with SSD were treated with antipsychotic medication, and it was shown that antipsychotic medication can interfere with visual perception among patients with schizophrenia [45].

It would be helpful to study the correlations between the illusion’s manifestation and the duration of medication use, disease duration, and specific disease symptoms.

## 5. Conclusions

Our study shows that the patients in the DDG were more susceptible to the O-K illusion, and the patients in the SSDG perceived the O-K illusion in a similar manner to the healthy volunteers. The patients who used BZD were more susceptible to the O-K illusion, and the BZD non-users perceived O-K figures in a similar manner to the healthy volunteers. A better understanding of the effect of benzodiazepines on visual perception could be helpful in finding potential biomarkers of benzodiazepine-related cognitive impairment.

## Figures and Tables

**Figure 1 medicina-61-00835-f001:**
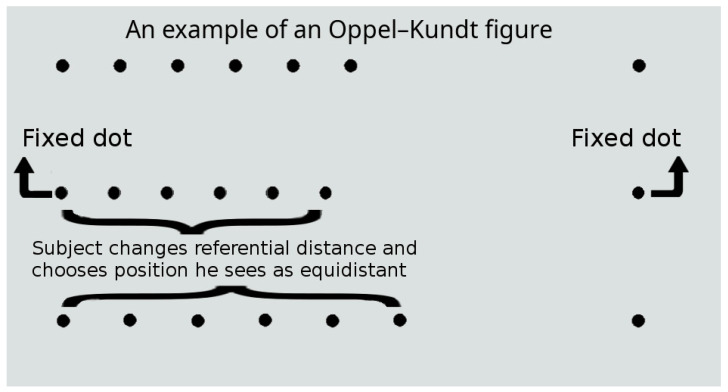
An example of how a subject performs an experiment.

**Figure 2 medicina-61-00835-f002:**
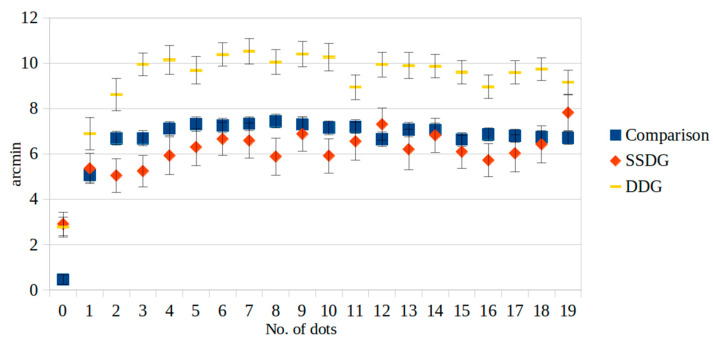
Susceptibility to the O-K illusion in patients with depressive (DDG) and schizophrenia spectrum disorders (SSDG).

**Figure 3 medicina-61-00835-f003:**
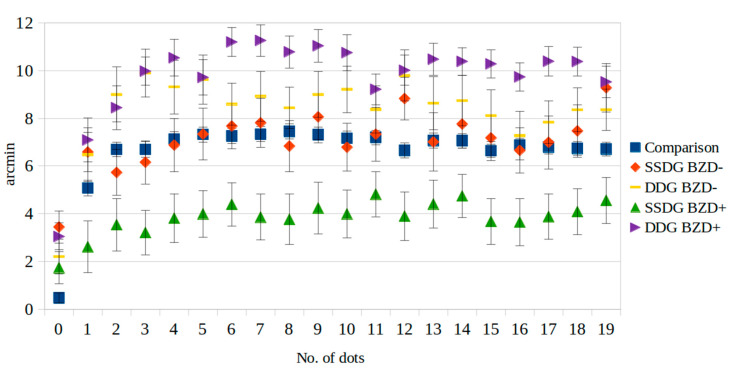
Susceptibility to the O-K illusion in patients with depressive disorders (DDG) and schizophrenia spectrum disorders (SSDG) relative to benzodiazepine use.

**Table 1 medicina-61-00835-t001:** Characteristics of the participants.

Characteristic	Comparison	SSDG	DDG	*p*
Age, y median (min.–max.)	28 (19–57)	35 (18–63)	31 (18–53)	0.426
Sex, male, n (%)	18 (51.4)	7 (36.8)	10 (62.5)	0.310
Benzodiazepine usage, n (%)	-	6 (31.6)	11 (68.8)	0.028
Disease duration, months median (min.–max.)	-	48 (1–360)	5 (1–72)	0.006
Diazepam equivalents, median (min.–max.)	-	5 (0–20)	0 (0–30)	0.892
Chlorpromazine equivalents, median (min.–max.)	-	200 (50–1000)	125 (0–900)	0.415

**Table 2 medicina-61-00835-t002:** Mean results in arcmin of each figure, with statistical significance across the patient groups.

No. of Dots	Comparison		SSDG		DDG		*p*
	Mean	SD	Mean	SD	Mean	SD	
0	0.47	4.04	2.92 *	7.00	2.78 *	5.68	<0.000
1	5.07	6.04	5.36	8.97	6.90 *	9.15	0.042
2	6.69	5.77	5.05 **	10.07	8.62 *	9.15	<0.000
3	6.69	6.21	5.25 **	9.37	9.95 *	6.44	<0.000
4	7.12	5.91	5.93 **	11.38	10.15 *	8.07	<0.000
5	7.32	5.89	6.31 **	11.06	9.69 *	7.65	<0.000
6	7.26	5.85	6.66 **	9.81	10.39 *	6.45	<0.000
7	7.33	5.87	6.59 **	10.45	10.53 *	7.15	<0.000
8	7.45	5.78	5.89 **	10.99	10.05 *	6.83	<0.000
9	7.31	6.15	6.88 **	10.17	10.40 *	7.04	<0.000
10	7.16	5.68	5.92 **	10.26	10.27 *	7.60	<0.000
11	7.20	5.66	6.56 **	11.21	8.95 *	6.89	0.013
12	6.65	5.77	7.31 **	9.56	9.95 *	6.91	<0.000
13	7.07	7.31	6.21 **	12.10	9.90 *	7.28	0.002
14	7.06	5.65	6.83 **	10.13	9.87 *	6.57	<0.000
15	6.64	5.38	6.10 **	9.74	9.60 *	6.65	<0.000
16	6.87	5.38	5.73 **	9.87	8.97 *	6.63	<0.000
17	6.80	5.50	6.03 **	11.16	9.59 *	6.49	<0.000
18	6.74	5.34	6.43 **	11.06	9.75 *	6.36	<0.000
19	6.71	5.32	7.83	10.63	9.16 *	6.64	<0.000

*—Statistically significant differences when compared to the comparison group. **—Statistically significant differences when compared to the DDG.

**Table 3 medicina-61-00835-t003:** Mean results of each figure in arcminutes for each figure by benzodiazepine use status, with statistical significance across the patient groups.

No. of Dots	Comparison		SSDG				DDG				*p*
			BZD+		BZD−		BZD+		BZD−		
	Mean	SD	Mean	SD	Mean	SD	Mean	SD	Mean	SD	
0	0.47	4.04	1.74	5.01	3.44 *	7.69	3.04 *	5.91	2.20	5.15	<0.001
1	5.07	6.04	2.61	8.06	6.59 **	9.11	7.09 **	9.70	6.47	7.90	0.002
2	6.69	5.77	3.53	8.25	5.73	10.74	8.45 **	9.61	9.00 **	8.14	<0.001
3	6.69	6.21	3.21 *	6.93	6.16 ***	10.17	9.98 *^,^**	6.15	9.90 *^,^**	7.10	<0.001
4	7.12	5.91	3.81 *	7.70	6.88	12.59	10.54 *^,^**	8.12	9.31 **	7.97	<0.001
5	7.32	5.89	3.99 *	7.36	7.33	12.24	9.71 *^,^**	7.85	9.63 **	7.27	<0.001
6	7.26	5.85	4.39 *	6.79	7.67 ***	10.76	11.20 *^,^**	6.41	8.60 **	6.25	<0.001
7	7.33	5.87	3.86 *	7.22	7.81 ***	11.42	11.26 *^,^**	6.97	8.92 **	7.33	<0.001
8	7.45	5.78	3.76 *	7.90	6.83 ***	12.02	10.79 *^,^**	7.05	8.44 **	6.11	<0.001
9	7.31	6.15	4.24	8.06	8.06	10.80	11.04 *^,^**	7.09	9.00 **	6.78	<0.001
10	7.16	5.68	3.99 *	7.45	6.78 ***	11.21	10.75 *^,^**	7.88	9.22 **	6.88	<0.001
11	7.20	5.66	4.82	7.04	7.33	12.58	9.21 **	6.84	8.37	7.05	0.010
12	6.65	5.77	3.90	7.60	8.83 **	9.96	10.01 *^,^**	6.65	9.80 **	7.52	<0.001
13	7.07	7.31	4.40	7.46	7.01	13.61	10.48 *^,^**	6.97	8.63	7.85	<0.001
14	7.06	5.65	4.74	6.76	7.76	11.21	10.38 *^,^**	6.11	8.74 **	7.44	<0.001
15	6.64	5.38	3.67 *	7.16	7.18	10.54	10.28 *^,^**	6.09	8.11 **	7.61	<0.001
16	6.87	5.38	3.64 *	7.35	6.65	10.70	9.74 *^,^**	6.24	7.27	7.19	<0.001
17	6.80	5.50	3.88 *	7.16	6.99	12.44	10.39 *^,^**	6.41	7.84 **	6.36	<0.001
18	6.74	5.34	4.08	7.18	7.47	12.28	10.38 *^,^**	6.24	8.35 **	6.46	<0.001
19	6.71	5.32	4.56	7.26	9.28 **	11.55	9.53 *^,^**	6.86	8.36 **	6.13	<0.001

*—Statistically significant differences when compared to the comparison group. **—Statistically significant differences when compared to the SSDG BZD+. ***—Statistically significant differences when compared to the DDG BZD+.

## Data Availability

The original contributions presented in this study are included in the article. Further inquiries can be directed to the corresponding author.

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
