# Peer review of "Variations in Oppel–Kundt Illusion Strength Among Depressive and Schizophrenia Spectrum Disorder Groups: Impact of Benzodiazepine Use"

_medicina, 2025, doi:10.3390/medicina61050835_

Round 1

Reviewer 1 Report

Comments and Suggestions for Authors

Hello, dear colleagues!

The article is devoted to a relevant topic and it is clear that you approached the solution of the problem thoroughly and methodically.

There are several questions about the work itself:

1. How did you calculate the sample size?

2. What were the exclusion criteria?

3. What were the limitations in working with the equipment? How did you exclude external light noise
4. Did you identify any adverse effects when using benzodiazepines in patients with schizophrenia?

5. It is advisable to update the list of references, since there are many sources that are more than 15 years old

Author Response

Dear Reviewer,

Thank you for your insights and kind comments. They have helped to enhance our article. We appreciate the time and commitment you gave in reviewing this article.

  1. How did you calculate the sample size?

We have calculated sample size with formula for comparing independent means. Based on equality of groups study design.

  1. What were the exclusion criteria?

The exclusion criteria for subjects were neurological or ophthalmological diseases, which could affect visual perception, motor system diseases causing an inability to perform the test, alcohol intake during the period of hospitalisation and a subject’s own refusal. I have marked exclusion criteria in the article with a color red. Page 2, lines 75-78.

  1. What were the limitations in working with the equipment? How did you exclude external light noise

Main limitation of the equipment was study subjects’ myopia: since figures were presented at 3,5 meters distance and not every subject had eyeglasses, some of them could not participate. Another limitation is that we can present only simple figures to the patients.  To avoid any light, we have used a totally dark room, without any windows.

4. Did you identify any adverse effects when using benzodiazepines in patients with schizophrenia?

No, we did not notice any adverse effects. Every patient was monitored by psychiatrist, we have collaborated with psychiatrists and patient and if the patient was not feeling well – we have postponed beginning of an experiment. This way we have achieved that experiments would be conducted in a best possible manner.

  1. It is advisable to update the list of references, since there are many sources that are more than 15 years old

Thank you for a kind remark, visual illusion studies have been done for a long time, but there is just a few labs working on this topic and not all recent studies are relevant to this specific type of visual illusion. This is why we chose to investigate specifically Oppel-Kundt illusion – it is less investigated, compared to other common physiological visual illusions.

Reviewer 2 Report

Comments and Suggestions for Authors

This study highlights the potential of visual perception differences as biomarkers for depression and the influence of BZD on visual processing. My questions and recommendations are:

  1. The document does not explicitly describe the methods used to exclude depression (Hamilton depression scale use for example) in healthy patients within the comparison group, which could be another limitation of the study.
  2. How did the authors calculate BZD equivalents?  Explain the methodology used for BZD conversion (The bibliography is incomplete, without pagination)!
  3. SDs are too high. The document does not explicitly discuss why authors use standard deviation (SD) versus standard error (SE).

Author Response

Dear Reviewer,

Thank you for your insights and kind comments. They have helped to enhance our article. We appreciate the time and commitment you have given to review this article.

1. The document does not explicitly describe the methods used to exclude depression (Hamilton depression scale use for example) in healthy patients within the comparison group, which could be another limitation of the study.

Each subject was evaluated with Brief Psychiatric Rating Scale by a certified psychiatrist. If there were any suspicion of a risk of any kind of mental disorder subject were not included in the comparison group. Thank you for your insight, I have marked changes red in the manuscript.  Page 2, lines 84-86.

2. How did the authors calculate BZD equivalents?  Explain the methodology used for BZD conversion (The bibliography is incomplete, without pagination)!

We have counted in accordance with table 1 provided in an Ashton manual, I have added pagination and marked changes in red. I have added additional explanation on results of patients with benzodiazepine use and marked it in red.  We did not find any dose-dependent correlation with benzodiazepine and visual perception; thus all benzodiazepine users were included independent of benzodiazepine dose. Page 11, reference No. 31.

3. SDs are too high. The document does not explicitly discuss why authors use standard deviation (SD) versus standard error (SE).

SD was chosen because it reflects the spread and variability of the data, which is important for describing the sample characteristics. Our goal was to highlight data distribution, which is why we used SD.

Reviewer 3 Report

Comments and Suggestions for Authors

The topic of the manuscript is quite relevant and interesting, this is due to the widespread prevalence of depressive disorders in the world. However, the authors should describe in more detail the possible application of their results in clinical practice, since the manuscript does not pay sufficient attention to this section. At the same time, this is an important aspect, due to which the relevance of the manuscript increases significantly. Also, in the materials and methods section, it is necessary to describe in more detail the level of severity of depression (subclinical level, clinical level ...). It is also known that depression is often accompanied by anxiety disorders, did the authors take this aspect into account? The authors need to add a table with a detailed clinical description of the groups of subjects (stage of the disease, the total duration of the disease, what drug therapy the patients received, dosages of drugs and their concentration in the blood, etc.). It is also not entirely clear how patients were selected based on the presence / absence of visual impairment, since the age range from 18 to 63 years suggests an age-related decrease in vision, and therefore the groups cannot be comparable. The authors need to expand the conclusion section, add clearer conclusions and a description of them with the possibility of application in clinical practice.

Author Response

Dear Reviewer,

Thank you for your insights and kind comments. They have helped to enhance our article. We appreciate the time and commitment you have given to review this article.

The topic of the manuscript is quite relevant and interesting, this is due to the widespread prevalence of depressive disorders in the world. However, the authors should describe in more detail the possible application of their results in clinical practice, since the manuscript does not pay sufficient attention to this section. At the same time, this is an important aspect, due to which the relevance of the manuscript increases significantly.

Risks of benzodiazepine use is widely known, and further research is needed to better understand impact of benzodiazepines on visual perception, but our findings could help with research of potential biomarkers of benzodiazepine cognitive impairment. Thank you for your kind remarks, changes in the manuscript were marked red. Page 8, lines 239 – 243.

Also, in the materials and methods section, it is necessary to describe in more detail the level of severity of depression (subclinical level, clinical level ...). It is also known that depression is often accompanied by anxiety disorders, did the authors take this aspect into account? The authors need to add a table with a detailed clinical description of the groups of subjects (stage of the disease, the total duration of the disease, what drug therapy the patients received, dosages of drugs and their concentration in the blood, etc.).

Thank you for this valuable insight, I’ve made improvements to the manuscript in accordance to your recommendations. I have marked it in red. Page 3, lines 93, 94, Table 1 and page 4, lines 136 to 142. During this study, we did not monitor drugs blood concentration, monitoring was not included in the study protocol.

It is also not entirely clear how patients were selected based on the presence / absence of visual impairment, since the age range from 18 to 63 years suggests an age-related decrease in vision, and therefore the groups cannot be comparable.

Thank you for your insight. We have excluded subjects with any ophthalmological disorder that would interfere with an experiment. Subjects with myopia were allowed to use prescribed eyeglasses. I have marked changes in red. Page 2, lines 75 to 79.

The authors need to expand the conclusion section, add clearer conclusions and a description of them with the possibility of application in clinical practice.

Thank you for your remark. I have included additional information on possible application in the conclusion and marked it red. Page 8, lines 239-242 and 259-261

Round 2

Reviewer 2 Report

Comments and Suggestions for Authors

I agree the modified version